# The Synergistic Benefit of Combination Strategies Targeting Tumor Cell Polyamine Homeostasis

**DOI:** 10.3390/ijms25158173

**Published:** 2024-07-26

**Authors:** Ting-Ann Liu, Tracy Murray Stewart, Robert A. Casero

**Affiliations:** 1Department of Biochemistry and Molecular Biology, Johns Hopkins Bloomberg School of Public Health, Baltimore, MD 21205, USA; tliu102@jh.edu; 2The Sidney Kimmel Comprehensive Cancer Center, School of Medicine, Johns Hopkins University, Baltimore, MD 21287, USA; tmurray2@jhmi.edu

**Keywords:** polyamine, polyamine transport, polyamine biosynthesis, polyamine catabolism, polyamine biosynthesis inhibitors, polyamine analogues, polyamine combination therapeutic strategies

## Abstract

Mammalian polyamines, including putrescine, spermidine, and spermine, are positively charged amines that are essential for all living cells including neoplastic cells. An increasing understanding of polyamine metabolism, its molecular functions, and its role in cancer has led to the interest in targeting polyamine metabolism as an anticancer strategy, as the metabolism of polyamines is frequently dysregulated in neoplastic disease. In addition, due to compensatory mechanisms, combination therapies are clinically more promising, as agents can work synergistically to achieve an effect beyond that of each strategy as a single agent. In this article, the nature of polyamines, their association with carcinogenesis, and the potential use of targeting polyamine metabolism in treating and preventing cancer as well as combination therapies are described. The goal is to review the latest strategies for targeting polyamine metabolism, highlighting new avenues for exploiting aberrant polyamine homeostasis for anticancer therapy and the mechanisms behind them.

## 1. Introduction

Polyamines are organic polycations that are found in all eukaryotes, plants, and most prokaryotes in millimolar concentrations [1]. The primary three forms of polyamines in mammals are putrescine, spermidine, and spermine (Figure 1) [2]. These molecules are essential for cell growth and survival and affect various cell processes. The protonation of amino groups at physiological pH facilitates electrostatic interactions between polyamines and negatively charged DNA, RNA, proteins, and phospholipids. This allows polyamines to function in regulating chromatin structure, DNA replication, transcription, translation, post-translational modification, ion-channel transport, free-radical scavenging, and membrane stability [3,4,5].

The major sources of polyamines in mammals are diet [6], microbial synthesis in the gut [7], and intracellular biosynthesis. Polyamine biosynthesis is initiated by the first rate-limiting enzyme ornithine decarboxylase (ODC), which catalyzes the decarboxylation of ornithine into putrescine (Figure 1) [8]. Further, the other two polyamines, spermidine and spermine, are generated by transferring propyl-amines from decarboxylated S-adenosyl-L-methionine (dcSAM) onto putrescine and spermidine to form spermidine and spermine, respectively. These reactions are catalyzed by the aminopropyl transferases spermidine synthase (SRM) and spermine synthase (SMS) [9]. The dcSAM is produced by a second rate-limiting enzyme, S-adenosyl-L-methionine decarboxylase (AMD1), which decarboxylates S-adenosyl-L-methionine (SAM). Polyamine catabolism is performed by spermidine/spermine *N*^1^-acetyltransferase (SSAT), which adds an acetyl group to spermine and spermidine to produce *N*^1^-acetyl-spermine and *N*^1^-acetyl-spermidine. These can then be exported out of the cell or serve as substrates for *N*^1^-acetylpolyamine oxidase (PAOX), which oxidizes them back to spermidine and putrescine, respectively. Spermine can also be directly oxidized back to spermidine, without the step of acetylation, via spermine oxidase (SMOX).

The critical roles of polyamines in normal cell function indicate the importance of maintaining the homeostasis of intracellular polyamine concentrations throughout the body. This occurs through coordinated biosynthesis, catabolism, and transport. Notably, polyamines regulate the cellular circadian clock, which in turn regulates polyamine synthesis enzymes [10]. In cancer, polyamine metabolism is frequently dysregulated, with elevated polyamine levels important for cancer cell proliferation and tumor progression. In fact, ODC was the first documented transcriptional target of the MYC oncogene [11], which broadly regulates metabolism, cell growth, and proliferation [12]. The self-regulatory nature of the pathway led to the development of polyamine analogues as a strategy for intervening in polyamine metabolism with an emphasis on inhibiting polyamine biosynthetic enzymes and promoting polyamine catabolic enzymes [13]. Since the development of the first polyamine analogue with potential to treat cancer, there have been many changes. The utility of polyamine biosynthesis inhibitors such as DFMO as antitumor strategies can be limited by multiple compensatory mechanisms that allow cancer cells to maintain high polyamine concentrations, such as increasing the uptake of exogenous polyamines by the polyamine transport system. The use of polyamine analogues such as bis(ethyl)norspermine (BENSpm) as monotherapies has been unsuccessful partially due to poor dosing schemes and dose-limiting toxicities [14,15]. Consequently, monotherapy strategies have not, thus far, achieved satisfactory clinical results. However, combination treatments may have greater potential for clinical use. The goal of combination strategies is to reduce the dose of each drug needed to achieve the effect, thereby reducing toxicities. This review will cover recent advances in the development of new strategies, including combination therapies, targeting polyamine metabolism for the prevention and treatment of cancer.

To target polyamine metabolism as a therapeutic strategy, an understanding of polyamine transport, synthesis, and degradation is essential, because resistance mechanisms may arise from any of these factors.

**Figure 1 ijms-25-08173-f001:**
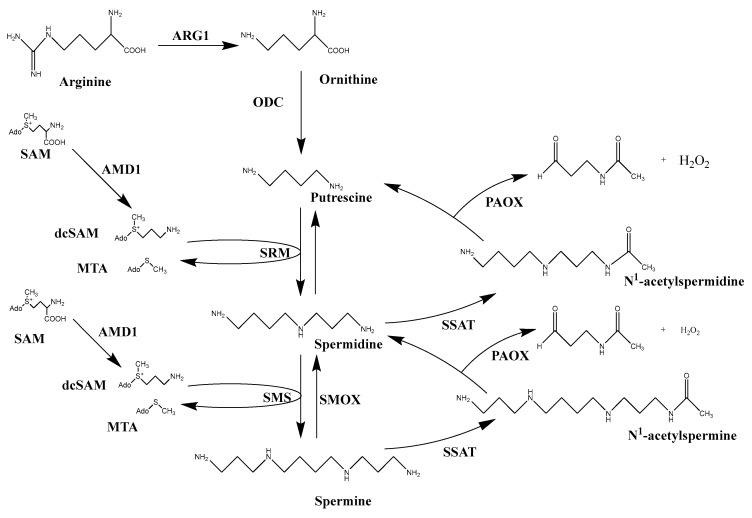
Polyamine synthesis and catabolism. Polyamine biosynthesis originates from the conversion of arginine to ornithine by arginase I (ARG1) in the urea cycle. Ornithine is then decarboxylated into putrescine by ornithine decarboxylase (ODC). S-adenosylmethionine (SAM) is decarboxylated by S-adenosylmethionine decarboxylase (AMD1) to form decarboxy-S-adenosylmethionine (dcSAM). The transfer of an aminopropyl group from dcSAM to putrescine by spermidine synthase (SRM) produces spermidine, while transfer of an aminopropyl group from dcSAM to spermidine by spermine synthase (SMS) produces spermine. Spermine is catabolized to spermidine either by direct oxidation via spermine oxidase (SMOX) or by the combined two-step pathway of spermidine/spermine *N*^1^-acetyltransferase (SSAT) acetylation and FAD-dependent polyamine oxidase (PAOX) oxidation. Spermidine can also be converted back to putrescine using the SSAT and PAOX pathway (Figure adapted from Casero 2019) [16].

## 2. Polyamine Transport

At physiological pH, polyamines naturally have a net positive charge. This prevents the passive diffusion of polyamines through the cell membrane. Polyamine transport is an energy-dependent process. However, the precise mechanism and molecules that are involved in the mammalian polyamine transport system remain poorly understood [17]. Three molecular models have been proposed for polyamine transport. The first model proposes a two-step process where, in mammals, a highly selective plasma membrane polyamine permease (PMPP) facilitates the initial influx of the polyamines into the cell through the power of electronegative membrane potential. Then, polyamines are rapidly internalized into acidic polyamine-sequestering vesicles (PSVs), which concentrate or disperse them throughout the cell as needed [18]. Internalization into these acidic PSVs requires a proton: polyamines exchange through vesicular proton-coupled polyamine antiporters [18]. The second model is selective for spermine and involves receptor-mediated endocytosis: spermine binds to cell surface heparin sulfate moieties in glypican 1 (GPC1), followed by the internalization of spermine into intracellular PSVs. Once inside the cell, spermine can be released through a nitric oxide-mediated oxidation process [19,20]. The third model proposes that putrescine is endocytosed through caveolin-1-dependent internalization in gastrointestinal tissues [21]. Even though there are many proposed mechanisms regarding polyamine transport, none of them have been proven to fully explain its components. It appears likely that eukaryotic cells do not utilize any single proposed transport mechanism for polyamines. Rather, all three of these proposed mechanisms may be involved, based on available biochemical data. Recently, there has been increased interest regarding the involvement of solute carrier transporters and P5B-type transport ATPases in polyamine transport. The solute carrier protein family 3 member 2 (SLC3A2) has been indicated in the concentration gradient-dependent exchange of putrescine with arginine [22]. ATP13A2 serves as a lysosomal polyamine exporter of endocytosed polyamines, and ATP13A3 functions in the import of polyamines [23,24,25]. Recently, another member of the P5B-ATPases, ATP13A4, was found to contribute to polyamine transport in mammalian cells [26], and another solute carrier, SLC18B1, was capable of in vitro polyamine transport through synthetic liposomes [27]. SLC18B1 also appears to contribute to polyamine transport in vivo, as indicated by significantly reduced polyamine concentrations in the brain tissue of SLC18B1 knock-out mice [28]. While there has been increased focus on understanding polyamine transport, future exploration is needed to understand the exact transport mechanisms and polyamine carriers that are involved, as this knowledge will provide valuable insight into drug delivery strategies targeting or exploiting the polyamine transport machinery for cellular access, as with the polyamine analogues. It is worth noting that there might not be only a single mechanism regulating polyamine transport but rather a cooperation of multiple mechanisms, and their roles may be tissue-specific and context-dependent.

## 3. Polyamine Biosynthesis

Polyamine homeostasis is highly regulated by interconnected biosynthetic, catabolic, and transport mechanisms under normal physiological conditions. In some pathologies, such as cancer, metabolic dysregulation such as increased polyamine biosynthesis and/or transport and decreased polyamine catabolism, alone or in combination, result in a sustained increase in intracellular polyamine concentrations, which correlates with increasing cancer cell proliferation, differentiation, and tumorigenesis. Elevated levels of polyamines have been associated with many types of cancer including breast, colon, prostate, and skin cancers [29,30,31].

In mammals, L-arginine can be derived from endogenous synthesis and the catabolism of protein found in meat, fish, dairy products, as well as nuts [32]. L-arginine is converted into the polyamine precursor ornithine by arginase (ARG1) during the urea cycle. Ornithine is decarboxylated by the rate-limiting enzyme ODC to form diamine putrescine (Figure 1). Active only as a homodimer, ODC has a relatively short half-life (10–30 min) [33]. It is regulated at every level, from transcription of the *ODC1* gene, to post-transcriptional processing and translation of mRNA, and altered stability of the protein [34,35,36,37]. The degradation of ODC is performed by regulatory proteins known as ODC antizymes (OAZs), which bind to ODC monomers and prevent them from forming an active dimer. There are three OAZ isoforms, OAZ1, OAZ2, and OAZ3, with different functions [38,39,40]. All members of the OAZ family can bind and inhibit ODC, but OAZ1 stimulates ODC degradation most effectively through 26S proteasomal degradation without ubiquitination [8], and OAZ2 is a potential compensatory factor for the loss of OAZ1 [41]. Antizyme inhibitor (AZI), which is a noncatalytic homolog of ODC, also regulates ODC. AZI has high sequence and structure homology to ODC, in which AZI binds to antizyme more tightly than ODC and can thus prevent ODC from binding with OAZs for degradation [42,43]. It is worth noting that the c-Myc/MycN and K-ras sarcoma virus (Ras) oncogenes, which are potential therapeutic targets for cancer, can also induce ODC [44,45]. L-arginine can also be converted to agmatine through a reaction catalyzed by the microbial enzyme arginine decarboxylase (ADC) (Figure 2) [46]. Agmatine acquired by mammalian cells can then undergo hydrolysis to urea and putrescine through agmatinase (AGMAT) [47]. This offers an alternative route to the biosynthesis of polyamines rather than the more widely recognized pathway involving ODC. 

**Figure 2 ijms-25-08173-f002:**
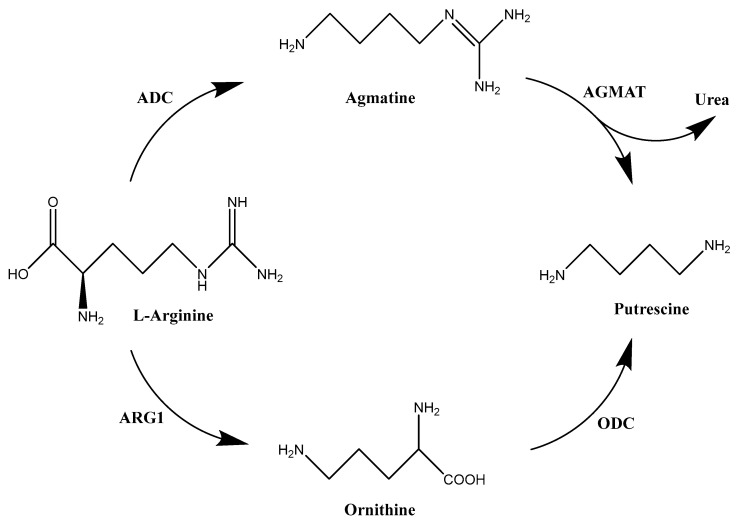
Alternative pathway of polyamine biosynthesis. Agmatine, an arginine intermediate, is synthesized from L-Arginine through microbial arginine decarboxylase (ADC). Acquired agmatine is then hydrolyzed to putrescine and urea through agmatinase (AGMAT) [48].

An aminopropyl moiety is necessary for the formation of the higher-order polyamines spermidine and spermine. This is provided by the second rate-limiting step of polyamine biosynthesis AMD1 (Figure 1) [49]. Putrescine enhances the activity of AMD1, thereby stimulating its own conversion to maintain higher levels of spermidine and spermine [50]. As with ODC, AMD1 has a relatively short half-life, and it is regulated by transcription, translation, and ubiquitin-dependent degradation when levels of polyamines are elevated [51,52].

As indicated above, spermidine and spermine are synthesized by two distinct aminopropyltransferases, SRM and SMS [53,54]. Spermidine is generated by the transfer of one aminopropyl moiety to putrescine by SRM. Spermine is generated by the transfer of another aminopropyl moiety to spermidine by SMS [55]. The activity of these two enzymes is regulated by the availability of their substrates. During the synthesis of spermidine and spermine, each reaction utilizes one molecule of dcSAM and produces 5′-deoxy-5′-(methylthio) adenosine (5′-MTA) [56]. 5′-MTA can be a potent inhibitor for both SRM and SMS. This inhibition by MTA can be prevented by rapid metabolism through the methionine salvage pathway and/or excretion of MTA from the cell. MTA is metabolized by 5′-methylthioadenosine phosphorylase (MTAP), forming adenine and 5′-methylthioribose-1-phosphate, which are further converted into ATP and methionine. Methionine further serves as a substrate for SAM synthesis through an ATP-dependent enzyme, methionine adenosyl-transferase 2 (MAT2), which transfers an adenosyl group to methionine to form SAM. Importantly, as the *MTAP* genomic locus is adjacent to tumor suppressor genes, it is frequently lost in cancers [57].

## 4. Polyamine Catabolism

Polyamines are interconvertible. The aminopropyl transferase reactions by SRM and SMS to form spermidine and spermine are irreversible, but polyamine catabolism can occur to maintain polyamine dynamics. There are two pathways for polyamine catabolism, which include amine oxidases. The first is a two-step pathway that starts with the *N*^1^-acetylation of spermidine or spermine by SSAT (Figure 1) [58,59]. These *N*^1^-acetylated polyamines can then be excreted out of the cell; however, they can also be substrates for PAOX. Flavin-dependent PAOX is a peroxisomal enzyme that preferentially oxidizes *N*^1^-acetyl-spermidine or *N*^1^-acetyl-spermine to putrescine or spermidine, respectively, with the formation of H_2_O_2_ and 3-aceto-aminopropanal (3-APP) as byproducts [60]. The second polyamine catabolic pathway is performed by SMOX, which has high homology to PAOX [61]. FAD-dependent SMOX activity directly catalyzes the oxidation of spermine to form 3-aminopropanal (3-AP), H_2_O_2_, and spermidine [62]. 3-Aminopropanal can be spontaneously converted to acrolein, another toxic byproduct that can cause damage [63]. These two catabolic pathways enable polyamine homeostasis by preventing excessive polyamine accumulation. However, both catalytic pathways have the caveat of generating considerable amounts of reactive oxygen species (ROS) and aldehydes, which can lead to DNA damage. The SMOX enzyme is localized in both the cytoplasm and nucleus [64], where it can be more destructive to DNA. PAOX is a peroxisomal enzyme, and the formation of H_2_O_2_ can be degraded rapidly in normal peroxisomes by catalase.

## 5. Polyamine Function

The functions of polyamines are multifaceted. Due to their polycationic nature at physiological pH, polyamines regulate gene expression by binding to negatively charged nucleic acids and proteins in chromatin. Thus, polyamines affect DNA, RNA, protein structure, ribosome function, and enzyme activity such as phosphorylation. Many transcription factors have been identified that respond to polyamine status. Specifically, the cellular circadian clock is affected by polyamines in such a fashion that, with aging, diminished polyamine levels result in clock disturbance [10]. Conversely, the oncogene c-Myc drives tumorigenesis in part by activating ODC to produce polyamines in proliferating cells [11]. 

Spermidine plays an essential role in a post-translational modification in which it is used as the substrate for the hypusine [*N*^8^-(4-amino-2-hydroxybutyl)lysine] modification. So far, only the eukaryotic translation initiation factor eIF5A has been reported to have this hypusination modification [65]. Hypusination involves two successive steps. First, the precursor form of eIF5A is modified by the attachment of the 4-aminobutyl group of spermidine by deoxyhypusine synthase (DHPS) to a specific lysine residue in the eIF5A precursor protein, generating the intermediate deoxyhypusine. Then, deoxyhypusine hydroxylase (DOHH) converts deoxyhypusine into hypusine to activate eIF5A (Figure 3) [66]. Hypusinated-eIF5A has been associated with cell growth and survival of eukaryotes through facilitating the translation of mRNAs with regions prone to ribosome stalling, such as poly-proline tracks [67]. 

**Figure 3 ijms-25-08173-f003:**
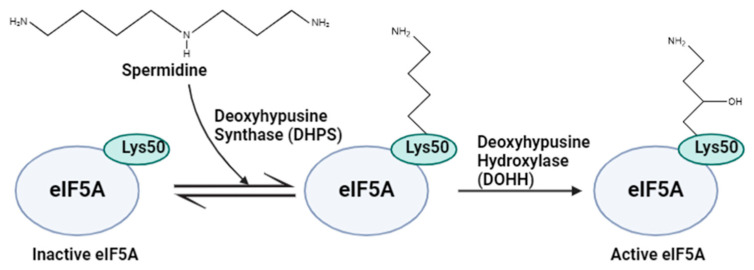
Schematic of eIF5A hypusination. Deoxyhypusine synthase (DHPS) transfers the 4-aminobutyl group from spermidine to a specific lysine residue (Lys50) in inactive eIF5A to yield the intermediate deoxyhypusine. Then, deoxyhypusine hydroxylase (DOHH) converts deoxyhypusine into hypusine to yield active eIF5A. (Figure adapted from Szepesi, Á., 2021) [68].

Polyamines also play a crucial role in epigenetics by altering the availability of methyl donors. Gene expression levels are regulated by the DNA methylation of CpG islands as well as the acetylation and methylation of lysine residues of histone tails [69]. AMD1 converts the methyl group donor SAM into dcSAM, limiting its availability for methyltransferase reactions. Hence, the ratio between dcSAM and SAM can affect the activity of DNA methylation [70]. Increased production of dcSAM can induce hypomethylation of cytosine residues in CpG islands and change the methylation state of lysine residues of histone tails [70]. In contrast, increasing polyamine intake inhibits AMD1 activity, which results in increased SAM and decreased dcSAM [71]. This abundance of SAM induces DNA methylation through the activation of DNA methyltransferase [72].

## 6. Polyamine Deprivation as an Anticancer Strategy: Arginine Deprivation and Agmatine

One of the recent adjuvant therapies for cancer that is being investigated is arginine deprivation therapy, which can be accomplished by inhibiting arginase [73]. Rather than its metabolism into ornithine by arginase, arginine is converted to nitric oxide (●NO) by nitric oxide synthase (NOS), with Nω-hydroxy-l-arginine (NOHA) as the intermediate product. NOHA can suppress the ability of arginase to produce ornithine [74], while nitric oxide is capable of suppressing ODC through S-nitrosylation of cysteine 360, located in the ODC active site [75]. As a result, blocking arginase activity could potentially impede polyamine biosynthesis by reducing ornithine production and favoring the production of nitric oxide, which in turn inhibits ODC. This is demonstrated by several studies in which treatments with arginase inhibitors such as rosuvastatin and L-norvaline can lower the cellular level of polyamines in tumor tissue, effectively inhibiting the growth of tumor cells and leading to apoptosis [76,77,78]. Conversely, overexpression of arginase can result in elevated levels of L-ornithine and putrescine, leading to the stimulation of tumor cell growth [77]. 

However, it is worth noting that arginine is also essential for the immune-mediated destruction of cancer cells. Also, long-term arginine deprivation therapy can cause cancer cells to produce arginine endogenously and compensate for the reduction in polyamine levels, thus decreasing their dependence on external sources of arginine [79]. This shifts the focus to an alternative pathway for polyamine biosynthesis that utilizes ADC to metabolize arginine into carbon dioxide and agmatine (Figure 2) [46]. Although the existence of a mammalian ADC is controversial, agmatine is a byproduct of bacterial ADC that can be acquired by mammalian cells via the gut microbiome and in certain foods [80]. Agmatine is then hydrolyzed into putrescine through the host agmatinase (AGMAT) [47]. However, there is an increased interest in using agmatine as an alternative for treating neoplasms. Agmatine has been known as an imidazoline receptor [81], and it is also involved in the homeostasis of polyamines. Previous studies have shown that agmatine regulates polyamine homeostasis through several effects: transport into hepatocytes by competing with putrescine [82], induction of SSAT leading to decreased polyamine levels [83], and reduction in ODC activity and protein [84]. Several studies also indicate agmatine as an anti-proliferative agent, in that it can inhibit cell proliferation by reducing the level of cellular polyamines [85,86,87,88]. A recent study in colorectal cancer suggests the potential use of recombinant ADC derived from E. coli as an antiproliferative strategy aimed at depleting arginine levels while promoting the generation of agmatine [89]. However, the toxic impact of agmatine on healthy cells must not be disregarded. Thus, additional research is required to fully understand the potential clinical use of agmatine. 

## 7. Polyamine Depletion as an Anticancer Strategy: Use of Enzyme Inhibitors

Multiple cancer types, including breast cancer and prostate cancer, are hallmarked by increased intracellular polyamine concentrations that enable continual cell growth [90]. Dysregulation of polyamine homeostasis is prevalent, resulting from upregulating polyamine biosynthesis, downregulating polyamine catabolism, increasing polyamine uptake, or any combination of these [16,91]. Therefore, it is feasible to assume that carefully designed molecules that target each aspect controlling polyamine access could be a rational approach for therapeutic benefit in dysregulated cell growth conditions such as cancer. 

One of the most successful and widely used inhibitors of polyamine biosynthesis is 2-difluoromethylornithine (DFMO, eflornithine; Table 1), which was developed in 1978. A synthetic analogue of the amino acid ornithine, DFMO is an enzyme-activated, irreversible ODC inhibitor [92]. Mechanistically, DFMO first competes with ornithine for binding to the ODC active site, where it forms a Schiff base with the aldehyde moiety of the pyridoxal phosphate cofactor. ODC then decarboxylates DFMO to form a highly reactive intermediate, which subsequently inactivates ODC through the formation of covalent linkages with either Cys360 (major) or Lys69 [93]. This results in the permanent inactivation of ODC. Importantly, when used as a treatment in cancer cells, DFMO generally results in cytostasis rather than cytotoxicity, as spermine pools are often preserved in spite of the depletion of putrescine and spermidine [94]. Therefore, DFMO can be used as an antiproliferative without causing cytotoxicity. Most recently, DFMO was FDA-approved for oral maintenance therapy in adult and pediatric neuroblastoma patients with an elevated risk of relapse and who have shown a minimal partial response to previous multiagent, multimodality therapy, which includes anti-GD2 immunotherapy [95,96]. It is worth noting that this marks the initial approval of an oncology drug that is based on an externally controlled trial (ECT) as the main clinical data to demonstrate significant evidence of efficacy [96]. 

An alternative to targeting ODC in polyamine synthesis is to focus on the second rate-limiting step AMD1. The first analogue shown to inhibit AMD1 was methylglyoxal bis(guanylhydrazone) (MGBG) (Table 1), a structural analogue of spermidine that competitively inhibits AMD1 and reduces the intracellular spermidine and spermine concentrations [97]. In the 1960s, MGBG was used as an anticancer drug. However, toxicities observed with MGBG treatment suggested that the dosage limitations may not be caused by polyamine depletion, but rather by other effects of the drug, such as anti-mitochondrial activity [98]. This mitochondrial toxicity has limited its use in further clinical trials [99]. 

Other AMD1 inhibitors were subsequently developed to reduce off-target toxicity. Second generation AMD1 inhibitors, 4-amidinoidan-1-one-2′-amidinhydrazone (SAM486A/CGP48664) (Table 1), and 5′(((z)-4-amino-2-butenyl)methylamino)-5′deoxyadenosine (AbeAdo) and its 8-methyl derivative Genz-644131 were created based on the structure of MGBG [100] and do not exhibit mitochondrial toxicity. SAM486A shows antiproliferative and antitumor effects, and it has undergone Phase I and II clinical trials [101,102]. It has also shown potential in combination with chemotherapies such as 5-fluorouracil, where cotreatment provided greater reduction in polyamines for therapeutic benefit [103].

For inhibiting higher polyamine synthesis, two compounds were synthesized to inhibit the respective aminopropyltransferase (spermidine or spermine synthase) efficiently and selectively. The design of S-adenosyl-3-thio-1,8-diaminooctane (AdoDATO) (Table 1) specifically inhibits spermidine synthase, and that of S-adenosyl-1,12-diamino-3-thio-9-azadodecane (AdoDATAD) (Table 1) specifically targets spermine synthase [104,105]. However, as the structures of both compounds contain primary amines, they will be subject to rapid intracellular metabolism by SSAT and SMOX. Therefore, they did not undergo further clinical trials. 

There are also potential therapeutic benefits in inhibiting polyamine catabolism. In response to inflammation and infection, which are associated with increased risk of cancer, expression and activity of the polyamine catabolic enzymes SSAT and SMOX are induced. These increases have been demonstrated in many cancer types and model systems, including lung, prostate, colon, stomach, and liver cancers [106,107,108]. Increases in either SSAT or SMOX cause the production of ROS. However, the ROS generated by SMOX specifically has been demonstrated to link inflammation and infection to carcinogenesis. SMOX expression is elevated in patients with prostate cancer and gastric cancer at all stages of disease [106,107]. Induction of SMOX activity results in elevated production of H_2_O_2_ and DNA damage. Inhibiting SMOX activity by *N*^1^, *N*^4^-di(buta-2,3-dien-1-yl)butane-1,4-diamine (MDL72527) (Table 1), a polyamine oxidase inhibitor, considerably decreases H_2_O_2_ generation, DNA damage, as well as tumor incidence [109]. These results indicate that inhibiting SMOX can be a potential target for chemoprevention for high-risk cancer patients. However, MDL72527 is not selective for SMOX but also inhibits PAOX [110]. Recently, the crystal structure of human SMOX was developed. The catalytic sites of human SMOX are different in terms of their substrate-binding domains, charges, and shape of the pockets, compared to PAOX [111]. This has facilitated the synthesis of novel compounds, such as JNJ-1289 and 2,11-Met2-Spm (Table 1), as potent inhibitors that are selective for SMOX over PAOX [111,112,113]. 

**Table 1 ijms-25-08173-t001:** Polyamine metabolism-targeting drugs. DFMO is an inhibitor of ODC. MGBG and SAM486A are inhibitors of AMD1. AdoDATO and AdoDATAD inhibit spermidine synthase and spermine synthase, respectively. MDL72527 inhibits both SMOX and PAOX. JNJ-1289 and 2,11-Met2-Spm are specific SMOX inhibitors.

Name	Structure	Reference
DFMO (Difluoromethylornithine)	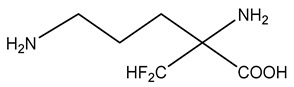	[92]
MGBG (Methylglyoxal (bis)guanylhydrazone)	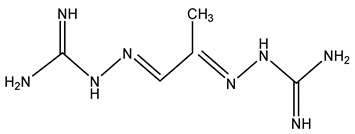	[97]
SAM486A (CGP48664)	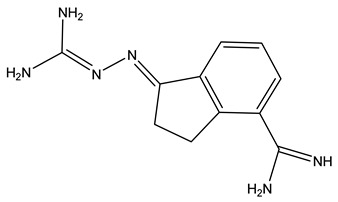	[100]
S-adenosyl-3-thio-1,8-diaminooctane (AdoDATO)	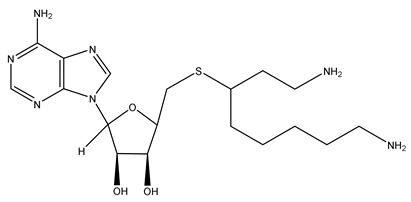	[104]
S-adenosyl-1,12-diamino-3-thio-9-azadodecane (AdoDATAD)	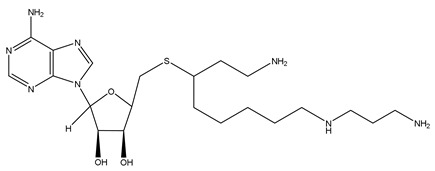	[105]
MDL72527	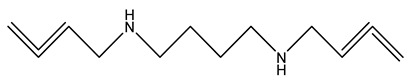	[109]
JNJ-1289	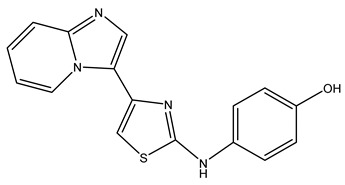	[111]
2,11-Met2-Spm	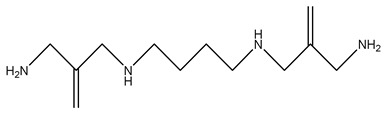	[113]

## 8. Synthetic Polyamine Analogues for Anticancer Treatment

The earliest polyamine-targeting strategies focused on the inhibition of polyamine biosynthetic enzymes; however, this provided limited clinical effects in reducing established tumors. When directly targeting inhibition of the polyamine biosynthetic enzymes, there was a major compensatory increase in polyamine transport to restore the depleted polyamine pools. Also, when ODC is inhibited, AMD1 activity is increased, and vice versa [114]. This results in a compensatory mechanism that facilitates cancer cells overcoming the effects of the inhibitors. Thus, with a breakthrough in understanding polyamine homeostasis, the development of polyamine analogues was used as a potential strategy for treating cancer. These polyamine analogues structurally resemble natural polyamines enough to enable competition for cellular uptake through the polyamine transport system, thereby acting as competitive inhibitors of the transport of natural polyamines. Also, the analogues downregulate polyamine biosynthesis through negative feedback mechanisms. Finally, the polyamine analogues differ from the natural polyamines enough to prevent their functioning as a natural polyamine replacement in cell growth and survival functions. Additionally, they are unable to serve as substrates for the polyamine catabolic enzymes [115]. 

The first analogues designed were symmetrical, terminally substituted bis(alkyl)polyamine analogues. The rationale for designing these analogues is that they are structurally similar to natural spermidine and spermine; however, the addition of bis(alkyl) groups to their primary amino termini provides protection from oxidation by amine oxidases. Examples of these analogues include *N*^1^, *N*^11^-bis(ethyl)norspermine (BENSpm) and *N*^1^,*N*^12^-bis(ethyl)spermine (BESpm) (Table 2) [116,117]. Mechanistically, these two analogues use feedback mechanisms to downregulate the polyamine biosynthetic enzymes ODC and AMD1, compete for polyamine transport, and dramatically induce the catabolic enzymes SSAT and SMOX, altogether resulting in rapid cellular polyamine depletion, increased ROS production, induction of apoptosis, and cancer cell death in sensitive cell types [118,119]. This can lead to tumor-selective cytotoxicity. In vitro and in vivo work has demonstrated that BENSpm can show considerable anti-tumor effects. Clinical trials were initiated with BENSpm but were not continued due to the emergence of a rare syndrome causing acute central nervous system toxicity [120,121]. However, it may be possible to utilize lower clinical doses of BENSpm in the future to reduce this toxicity, which is BENSpm-specific, highlighting the potential for its use when combined with other therapeutic strategies [118].

Expanding upon this self-regulating criteria, a second generation of polyamine analogues was developed that comprised unsymmetrically substituted polyamine analogues, such as *N*^1^-(cyclopropylmethyl)-*N*^11^-ethylnorspermine (CPENSpm) and *N*^1^-propargyl-*N*^11^-ethylnorspermine (PENSpm) (Table 2). Both CPENSpm and PENSpm treatment result in induction of SSAT and cytotoxicity [122]. This result demonstrated that minor changes in the structure of the polyamine backbone of the analogue can significantly alter the biological responses. It also demonstrated that the addition of reactive functional groups to the polyamine backbone could improve affinity of the analogues for the polyamine transport system, facilitating cellular entry and providing leads for the synthesis of many subsequent compounds, including those incorporating a polyamine moiety as a targeting vector [123,124,125]. 

The third variant of the bis(alkyl)-polyamine analogues introduced rotational restriction at the central carbons of the polyamine chain. These compounds include PG-11047 and PG-11093 (Table 2) [126,127]. PG-11047 is based on the structure of BENSpm but includes a cis double bond between the two central carbons of the 4-carbon methylene bridge. This double bond conformationally restricts the structure, the goal of which is to reduce the off-target effects seen in clinical trials for BENSpm. PG-11093 is a BEHSpm (bis(ethyl)homospermine) analogue that includes a cyclopropyl ring at the two central carbons. These conformationally restricted analogues increase the anti-proliferative activity and reduce the overall non-specific toxicity associated with the original BENSpm and BEHSpm compounds. In vitro, PG-11047 inhibits the growth of lung, breast, and colon cancer cell lines [126,128,129,130]. As with the original bis(ethyl) spermine analogues, PG-11047 showed differential sensitivity between small cell lung cancer and non-small cell lung cancers, with non-small cell lung cancer showing dramatically greater induction of polyamine metabolism in association with greater polyamine depletion and cytotoxicity [126]. PG-11047 was found to be well tolerated in Phase I clinical trials as a single agent and in combination with other standard-of-care chemotherapeutics in Phase Ib clinical trials [131]. 

Another polyamine analogue family is known as the oligoamines. The rationale for oligoamines was to increase the affinity of these polyamine analogues to negatively charged DNA by increasing the number of protonatable nitrogens [132]. The oligoamine PG-11144 (Table 2) showed anti-tumor benefits in breast cancer models in vivo and in vitro [133]. Oligoamines have higher potential to aggregate DNA, and PG-11144 has significant epigenetic effects resulting in gene expression changes. Some oligoamines, especially PG-11144 also inhibit LSD1 [134]. Lysine-specific demethylase 1, LSD1, is capable of extensively suppressing the expression of genes by catalyzing the demethylation of H3K4me2/me1 through an oxidative process, and several studies suggested elevated levels of LSD1 are observed in certain types of human cancers [135,136]. Numerous studies reported that the inhibition of LSD1 by distinct compounds led to the re-expression of several abnormally silenced tumor suppressor genes, providing evidence for the potential of targeting LSD1 for therapeutic treatment [137,138,139]. The structural analysis of LSD1 shows that its FAD-binding domain is highly homologous to SMOX and PAOX [135]. By inhibiting LSD1 activity, these analogues act as epigenetic regulators, reactivating aberrantly silenced genes important in tumorigenesis, such as SFRP2 and the Wnt signaling pathway [134]. 

**Table 2 ijms-25-08173-t002:** Polyamine analogues. Polyamines analogues can be organized into different groups based on their substitutions. BENSpm and BESpm are symmetrically substituted analogues. CPENSpm and PENSpm are asymmetrically substituted analogues. PG-11047 and PG-11093 are conformationally restricted polyamine analogues. PG-11144 is an example of an oligoamine analogue. SBP-101 (ivospemin) is a spermine analogue that is structurally similar to BENSpm.

Name	Structure	Reference
BENSpm	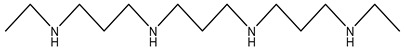	[116]
BESpm	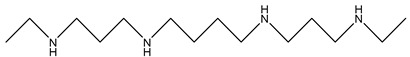	[117]
CPENSpm	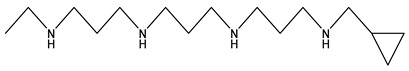	[122]
PENSpm	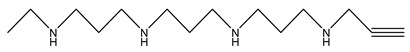	[122]
PG-11047	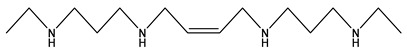	[125]
PG-11093	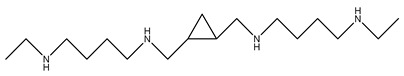	[126]
PG-11144	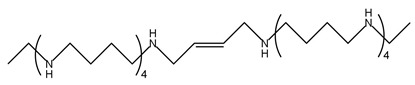	[132]
SBP-101 (ivospemin)	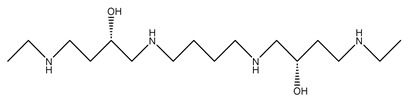	[140]

One spermine analogue currently under investigation is SBP-101, or ivospemin (diethyl dihydroxyhomospermine) (Table 2) [140]. Even though it is structurally similar to BENSpm, SBP-101 shows modest induction of polyamine catabolism, but stronger repression of ODC activity. During Phase Ia/b clinical trials, the combination of SBP-101 with FDA-approved, standard chemotherapeutics nab-paclitaxel and gemcitabine inhibited tumor growth in pancreatic cancer patients [141]. Currently, patients are being recruited into the multi-center ASPIRE Phase II/III clinical trial, which involves the combination of SBP-101 with nab-paclitaxel and gemcitabine for the treatment of metastatic pancreatic ductal adenocarcinoma (NCT05254171) [142]. In vitro, SBP-101 treatment reduces viability of lung, pancreatic, and ovarian cancer cells, and in the syngeneic VDID8+ murine ovarian cancer model, SBP-101 extends median survival in association with decreased intracellular polyamine pools [143,144]. 

## 9. Polyamine Transport Inhibitors

While the direct mechanisms for polyamine transport are not fully understood, there are available inhibitors for targeting polyamine transport to regulate polyamine uptake. Trimer44NMe (Table 3) is one such polyamine transport inhibitor that contains three polyamine side chains attached to a 1,3,5-substituted benzene core [145]. It is understood to block natural polyamine uptake in mammalian cells. Another polyamine transport inhibitor, AMXT1501 (Table 3), was developed with a lysine–spermine conjugate structure [146,147]. However, as with all polyamine-based compounds, polyamine transport inhibitors that contain aminopropyl termini will potentially be oxidized by polyamine oxidases that are present in the fetal bovine serum (FBS) used for cell culture [148]. Therefore, they are easily degraded in vitro, potentially affecting compound potency. 

As with the polyamine analogues described above, one strategy to prevent this route of degradation uses N-alkylation of the primary amine, converting it into a secondary amine, and providing extra metabolic stability in culture [149]. However, these transport inhibitor conjugates failed to target the polyamine transport system. Therefore, to increase affinity for and targeting of the polyamine transport system, another new approach was developed. Nitrogen-containing macrocycle rings containing additional charges and hydrogen bonding partners were conjugated with polyamines. In this case, secondary amines were maintained at one terminus, while the other terminal end with macrocycles could target the polyamine transport system. These new polyamine transport inhibitors showed potent polyamine transport inhibition [150]. 

## 10. Synergistic Polyamine Combination Therapeutic Strategies

Single-agent cancer therapeutics are often unable to achieve satisfactory clinical effects due to acquired drug resistance, tumor heterogeneity, and compensatory mechanisms. For instance, repression of polyamine biosynthesis leads to an increase in extracellular polyamine entry to replenish intracellular polyamine pools. Now, the focus has moved from single agents forward to combination therapies in an effort to offer more targeted drug delivery, reduce effective doses, and elicit longer-lasting clinical responses. 

Nanotechnology can be used to provide low-toxicity and high-efficacy agents for cancer therapy [151]. One of the recent advances is using polymeric nanotechnology to produce polyamine-based, polycationic nanocarriers. The rationale was to affect polyamine homeostasis while serving to deliver therapeutic nucleic acids, antitumor polyamine analogs, or other kinds of antitumor drugs. Targeting the polyamine biosynthetic pathway genes through RNA-based therapeutics shows great potential as a cancer treatment approach [152]. These include RNA interference (RNAi), siRNA, microRNA mimics and inhibitors, antisense oligonucleotides (ASO), RNA aptamers, ribozymes, and long non-coding RNA (lncRNA) [153]. There are several studies on the application of RNAi and siRNA to target polyamine biosynthetic genes in gastric cancer and breast cancer cell lines, showing dose- and time-dependent downregulation of ODC, SRM, and AMD1 genes, with inhibition of cell growth and induction of cell death through apoptosis [154,155,156]. There is also a study using a combination of agmatine, siRNA, and bovine serum albumin to synergistically inhibit cancer cell proliferation. The rationale was to synthesize a pH-sensitive cationic poly(agmatine), consisting of a poly(4-vinyl benzaldehyde) main chain linked with agmatine through a Schiff base. This poly(agmatine) can compact with siRNA targeting AMD1 (siAMD1) into a nanoparticle core, which is then coated with bovine serum albumin to enhance its stability in circulation [157]. In this case, the nanoparticle can co-deliver combinational therapy to have synergistic effects: agmatine and siAMD1 can be released in a pH-dependent manner, working together to inhibit polyamine biosynthesis and inhibit liver cancer cell growth [157]. 

In another promising study, Yathindranath et al. examined the effectiveness of ionizable nanoparticles containing SAT1-targeted siRNA in glioblastoma cells [158]. High SAT1 expression in glioblastoma cells is associated with resistance to chemotherapy and radiation. However, these experiments indicated that reduction in SAT1 expression increased the sensitivity to both chemotherapeutic agents and radiation in several representative glioblastoma lines. Consequently, these results suggest that combining inhibition of SAT1 with standard-of-care treatment in glioblastoma may be a strategy for increasing the responsiveness of this devastating tumor type. 

Another use of nanotechnology is the synthesis of biodegradable polycation prodrugs linked with polyamine analogues. Two of these nanopolyamines are Nano11047 (Table 4) and DSS-BEN (Table 4), which are synthesized from PG-11047 and BENSpm, respectively, and are linked by a bis(2-hydroxyethyl) disulfide (BHED) linker [159,160]. When entering into a reductive intracellular environment, this disulfide linker is readily cleaved to release unmodified BENSpm or PG-11047. The structures of these nanopolyamines are dendritically branched, and, similar to other known cationic dendrimers, they are transported into cells by endocytosis, followed by endosomal escape and self-immolation to their parent compounds. These nanopolyamines were designed to overcome potential resistance to polyamine analogues through the downregulation of polyamine transport. In this case, nanopolyamines would be accumulated in cancer cells through general endocytosis, instead of the polyamine transport system. Both DSS-BEN and Nano11047 treatment induced polyamine catabolism and had anticancer effects both in vitro and in vivo [160]. Additionally, a promising anticancer strategy includes combining anticancer drugs with therapeutic microRNA (miRNA), and these nanopolyamines can serve as a desirable delivery system. The simultaneous delivery of DSS-BEN with miR-34a, which targets p53 [161], suppressed polyamine biosynthesis, induced polyamine catabolism, and enhanced cytotoxicity in colon cancer cells, with improvement in antitumor effects in vivo [162]. However, recently, it was found that the internalization mechanism used by these nanopolyamines shares an unidentified, critical component with the polyamine transport system of the natural polyamines and their parent analogues [163]. 

Finally, fluorescent nanoparticles have been designed with functional abilities allowing polyamine detection and quantification while also serving to consume polyamines, resulting in tumor cell death. This and other innovative strategies involving supramolecular polyamine conjugates and nanoparticles were recently reviewed elsewhere, to which we refer readers [164].

When using polyamine metabolism as an anticancer strategy, cells can import polyamines from extracellular sources, such as dietary intake or intestinal microbial flora, via the bloodstream to conquer polyamine depletion. This can help cancer cells overcome or compensate during treatment with DFMO or other polyamine depletion drugs. To overcome this compensation, one approach is to combine DFMO treatment with a reduction in dietary polyamine intake [165]. However, the gut flora is a rich source of polyamines, and polyamines are present in essentially all foods [166]. Thus, combining drugs that target both the import and biosynthesis of polyamines has become a feasible strategy for sustained polyamine restriction, now known as “polyamine blocking therapy (PBT)”. Many studies have demonstrated an antitumor effect from combining treatment with the polyamine uptake inhibitor AMXT1501 with DFMO in both in vitro and in vivo models, such as those of breast, prostate, and ovarian cancers [167,168]. When given as monotherapy, both DFMO and AMXT1501 showed limited effects as the rescue pathway would occur. However, giving DFMO and AMXT1501 together effectively targeted several types of cancer cells and delayed tumor formation. Based on these results, a Phase Ib/IIa clinical trial (NCT05500508) using DFMO and AMXT1501 as a combination treatment is undergoing evaluation in patients with advanced solid tumors. In a similar study, DFMO in combination with other polyamine transport inhibitors (Trimer44NMe or MQT1426), both in vitro and in vivo, also showed increases in the survival of tumor-bearing mice, with reduced tumor growth [169,170]. 

Non-steroidal anti-inflammatory drugs (NSAIDs) are commonly used as anti-inflammatory agents [171]. NSAIDs have been shown to reduce the risk of cancer and may extend cancer patient survival [172]. NSAIDs, including sulindac, aspirin, and indomethacin, induce SSAT expression and polyamine export [173,174,175]. It was shown that the induction of apoptosis and inhibition of cell proliferation following sulindac exposure was polyamine-dependent, occurring via SSAT induction and intracellular polyamine content depletion [174]. These findings suggested that NSAIDs could play a potential role in combination therapy through upregulating SSAT. The combination of DFMO and sulindac significantly inhibited adenocarcinoma via anti-inflammatory, proapoptotic, and anti-proliferative effects beyond those of each single agent alone or NO-sulindac (Nitric-oxide releasing sulindac) plus DFMO. NO-sulindac was designed to reduce the gastric toxicity associated with NSAIDs [176], with the rationale of delivering NO to the site of damage [177]. However, the effect of combining DFMO and NO-sulindac was modest compared to DFMO alone. One possible reason might be that NO-sulindac acts on polyamine metabolism differently from sulindac, which induces SSAT [174]. Overall, these results demonstrated that the combination of DFMO and sulindac modulates polyamine metabolism synergistically [178]. DFMO was also tested with celecoxib and the chemotherapies cyclophosphamide and topotecan in a Phase I trial of heavily pretreated patients (NCT02030964) [179]. Indeed, combining DFMO with celecoxib enhanced antitumor activities across preclinical models [180]. Combinations of DFMO with other NSAIDs, such as piroxicam, indomethacin, and aspirin, have also been tested, and their efficacies were reported [181,182,183].

Chemoprevention strategies are influential in suppressing, reversing, and preventing the carcinogenic advancement to invasive cancer. However, current chemotherapy causes side effects without a complete cure, with risks of cancer recurrence and chemotherapeutic drug resistance. Thus, alternative strategies for cancer prevention and treatment are being explored, such as low-dose combinations of agents to improve anticancer efficacy and eliminate unwanted toxicities. Combining chemotherapeutic agents with polyamine analogues or other drugs targeting the polyamine metabolic pathway has been explored in multiple studies. These chemotherapeutic agents include paclitaxel, docetaxel, 5-fluorouracil, cis-diaminechloroplatinum (II), fluorodeoxyuridine, and vinorelbine [184]. In a Phase Ib study, PG-11047 combinational treatment with bevacizumab, erlotinib, cisplatin, or 5-fluorouracil in patients with advanced solid tumors showed that it is safe to administer PG-11047 in these combinations, and that the combinations may provide added therapeutic benefit [185].

Emerging data imply that polyamines play an important role in immune system regulation and function. This offers opportunities to use polyamines as a target for immune therapy. Epigenetic therapies such as DNA methyltransferase and histone deacetylase inhibitors (DNMTi and HDACi) have been shown to induce type I IFN signaling, which reduces an immune-suppressive microenvironment [186]. 5-Azacytidine (AZA) is a DNA methyl transferase inhibitor that can cause the re-expression of hypermethylated tumor suppressor genes in cancer cells [187]. The triple combination of the DNMTi AZA, the HDACi MS275, and the immune checkpoint inhibitor α-PD-1 enhanced the antitumor effects and increased overall survival [186]. Treatment of an in vivo model of ovarian cancer with DFMO and 5-azacytidine amplified the accumulation of pro-inflammatory M1 macrophages, T cells, and natural killer cells, and abated the immunosuppressive tumor microenvironment, including decreasing the population of M2-polarized macrophages [188]. The result was improved overall survival in mice receiving the combination as compared to single-agent treatment.

Furthermore, it has been shown that the anticancer effects of combining DFMO and polyamine transport inhibitors depend on the immune system, as they both deplete polyamines within tumor cells while also relieving polyamine-mediated immunosuppression in the tumor microenvironment, thereby activating T cells [189]. The therapeutic efficacy of polyamine blockade therapy can be completely abrogated by the depletion of CD8+ T cells [190]. Similarly, the antitumor effect of DFMO observed in intact immunocompetent mice was also abrogated in immunodeficient Rag1^−/−^ mice [191]. This demonstrated the antitumor effects of DFMO in modulating immune responses. Therefore, there is also a possibility of combining polyamine-blocking therapy with checkpoint immunotherapy using anti-PD1 or anti-PDL1. DFMO and α-PD-1 combination treatment showed synergistic effects by increasing the survival and activity of intratumoral CD8+ T cells [192]. 

One of the functions of polyamines is that they are involved in DNA repair by inducing the enzymatic activity of PARP1 [193]. Thus, elevated polyamines may lead to resistance to chemotherapy that targets DNA, such as DNA-damaging platinum drugs. Indeed, cotreatment with DFMO and PARP inhibitors, such as rucaparib, can enhance the cytotoxicity of the chemotherapeutic agent cisplatin [194]. 

Several studies have shown that combined treatments of bis(ethyl) polyamine analogues with chemotherapeutic drugs such as cisplatin and oxaliplatin provide superior anticancer effects compared to the chemotherapeutic alone [195,196,197]. These cotreatments enhance the translation and stability of SSAT protein, allowing polyamine pool depletion and further implicating SSAT induction as an anticancer therapeutic target. Recently, the combinations of ivospemin (SBP-101) with common chemotherapeutic agents such as gemcitabine, topotecan, paclitaxel, and docetaxel were examined in platinum-resistant ovarian cancer cell lines [198]. The results demonstrated increased responses to each of the four agents with the addition of ivospemin. However, in vitro, only combinations of gemcitabine or topotecan with ivospemin decreased the viability of ovarian cancer cells compared to ivospemin alone. In a murine ovarian cancer model, the combination of ivospemin with gemcitabine and topotecan also increased the survival of the mice compared to ivospemin monotherapy. These results emphasize the clinical potential of using ivospemin in combination with standard chemotherapy [198]. 

## 11. Future Perspectives and Conclusions

Increasing intracellular polyamine concentrations have been demonstrated in several cancer cell types, such as prostate, lung, ovarian, and breast cancers. Urinary levels of *N*^1^, *N*^12^-diacetylspermine have been identified as a biomarker for non-small cell lung and ovarian cancer [199,200]. Thus, the development of therapies targeting polyamine homeostasis is important. These include inhibitors of polyamine biosynthesis and polyamine transport, and inducers of polyamine catabolism as cancer-drug candidates. Due to the complexity and heterogeneity of tumors, polyamine compensation, and acquired resistance to drugs, monotherapies for cancer often face limitations. A combination of polyamine depletion and polyamine transport as a blocking strategy or a combination of polyamine depletion and chemotherapy may have greater efficacy. In conclusion, polyamine-based antitumor treatments are a rational approach to cancer therapy. However, more studies are necessary to define efficacious combinations to improve clinical response.

## Figures and Tables

**Table 3 ijms-25-08173-t003:** Polyamine transport inhibitors. Polyamine transport can be blocked by polyamine transport inhibitors (PTI) including Trimer44NMe and AMXT1501.

Name	Structure	Reference
Trimer44NMe	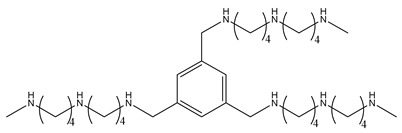	[145]
AMXT1501	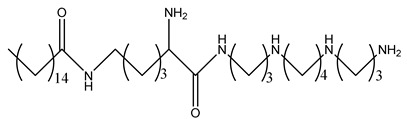	[146]

**Table 4 ijms-25-08173-t004:** Nanopolyamines. DSS-BEN is a nanoparticle compound synthesized from BENSpm. Nano11047 is a nanoparticle compound synthesized from PG-11047.

Name	Structure	Reference
DSS-BEN	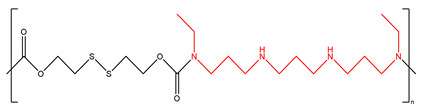	[162]
Nano11047	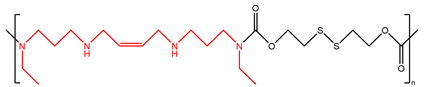	[160]

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
