# Peer review of "The Synergistic Benefit of Combination Strategies Targeting Tumor Cell Polyamine Homeostasis"

_ijms, 2024, doi:10.3390/ijms25158173_

Round 1

Reviewer 1 Report

Comments and Suggestions for Authors

Review of the Manuscript: "The synergistic benefit of combination strategies targeting tumor cell polyamine homeostasis"

Introduction

The manuscript titled "The synergistic benefit of combination strategies targeting tumor cell polyamine homeostasis" presents a review of the mechanisms of polyamine metabolism and therapeutic strategies aimed at modulating these pathways for cancer treatment. However, the work shows significant shortcomings in considering the most recent discoveries and new molecules studied in the field. Overall, the content needs a proper update or integration of the most current information.

Incomplete Consideration of Recent Discoveries

While the manuscript provides a detailed discussion of the synthesis and catabolism mechanisms of polyamines, it lacks an in-depth treatment of new molecules and recent scientific discoveries that have influenced the field in recent years.

For example, recent studies on the selective inhibition of specific enzymes such as spermidine/spermine N1-acetyltransferase (SSAT) are not mentioned. These studies have provided new insights into targeted cancer therapies that could be crucial for a comprehensive review.

Additionally, new developments in polyamine transporters like ATP13A2 and SLC18B1 are also omitted. These transporters have been identified as significant in the regulation of polyamine homeostasis and their role in cancer therapy is an emerging area of interest.

The review ignores recent progress in the use of nanomaterials for polyamine targeting, an interesting and developing topic that deserves at least a mention. Nanomaterials have shown potential in enhancing the delivery and efficacy of polyamine-targeting drugs, representing a significant advancement in the field.

Lack of Inclusion of New Molecules

While the manuscript provides an overview of traditional molecules used to inhibit polyamine metabolism, such as DFMO and BENSpm, it fails to include new molecules that have shown efficacy in recent studies.

For instance, the recently developed specific inhibitors of SMOX (e.g., JNJ-1289) and new polyamine analogs with innovative chemical structures that overcome the limitations of previous agents are not discussed. These new developments are critical for understanding the current landscape of polyamine-based therapies.

Much of the content and figures presented in the manuscript resemble those already present in previous works, without developing new emerging metabolic lines and without providing significant new contributions or innovative interpretations. This repetition indicates a lack of originality and fails to advance the existing knowledge in the field.

Conclusion

In summary, the reviewed manuscript requires substantial revision to include the most recent scientific discoveries and new molecules studied in the field of polyamine metabolism modulation. In its current form, the work appears to be a repetition of previous research and lacks innovation and updates. A thorough revision integrating new scientific evidence could significantly improve the quality and usefulness of the manuscript for the scientific community.

Reviewer 2 Report

Comments and Suggestions for Authors

Comments for the authors

In their manuscript Liu et al provide a comprehensive review concerning potential therapies for the treatment of malignancies, based on principles to influence polyamine metabolism. Importantly, the authors focuse on combination strategies, providing an interesting list of more or less proteomic possibilities.

Unfortunately, the increasing importance of non-coding RNA is ignored so far. This negligence should be corrected in the final manuscript.

Reviewer 3 Report

Comments and Suggestions for Authors

The manuscript titled "The synergistic benefit of combination strategies targeting tumor cell polyamine homeostasis" presents a review of the mechanisms of polyamine metabolism and the therapeutic strategies aimed at modulating these pathways for cancer treatment. However, it shows significant shortcomings in considering the most recent discoveries and new molecules studied in the field, lacking an in-depth discussion of the scientific innovations that have influenced the sector in recent years. Moreover, it revisits topics that have already been extensively covered in previous reviews.

Critical Points:

·       The review ignores recent progress in the use of nanomaterials for polyamine targeting, an interesting and developing topic that deserves at least a mention. Nanomaterials have shown potential in improving the delivery and efficacy of polyamine-targeting drugs, representing a significant advancement in the field.

·       Among the significant gaps in the manuscript is the absence of any reference to agmatine and its role in tumors. Agmatine, a derivative of arginine decarboxylation, has been recently studied for its potential impact on tumor growth modulation and response to antitumor therapies. Completely ignoring this molecule represents a serious omission, considering the implications it could have in the context of polyamine metabolism and therapeutic strategies for cancer.

·       The manuscript does not adequately address the role of arginine in tumor biology, particularly its conversion into ornithine and the subsequent synthesis of polyamines. The role of arginine as a precursor to polyamines and its potential as a therapeutic target is a critical aspect missing from the discussion. The metabolism of arginine to produce ornithine or agmatine, and thus the choice of metabolic switch, could be a crucial factor in the synthesis of polyamines and tumor growth.

In summary, the manuscript requires substantial revision to include the most recent scientific discoveries and new molecules studied in the field of polyamine metabolism modulation. In its current form, the work appears to be a repetition of previous research and lacks innovation and updates. A thorough revision integrating new scientific evidence could significantly improve the quality and usefulness of the manuscript for the scientific community.

Reviewer 4 Report

Comments and Suggestions for Authors

Liu et al. have written a thorough and informative review that is highly appropriate for this special issue and will be welcomed by the field.  It includes a clear and concise introduction to the biochemistry and metabolism of polyamines followed multiple sections that detail diverse strategies to target this biochemistry.  A few suggestions for improvement are offered below.

1.     The authors deftly navigate a large body of literature that has been frequently reviewed by others.  Increased citation of such reviews would be welcome to guide additional reading for those that are new to the field (e.g. dedicated reviews on ODC or AMD1 inhibitors).

2.     Figure 2 would be more informative if it showed the molecular details for the hypusination of the lysine residue in eIF5A rather than a simplified cartoon.

3.     The DFMO/neuroblastoma section near line 253 is very brief given that this approach is FDA approved.

4.     Table 1 and others should be refined to scale the structures to equalize proportions and enhance visibility (e.g. SAM486 is huge compared to the others). This would be facilitated by removing the large amount of empty white space to the right of the structures.  Some details of mechanisms of inhibition/action would also be helpful.  Why not inclusion of SMOX inhibitors?

5.     The section including lines 366-370 could be greatly enhanced by increased detail of the “significant epigenetic effects” and a definition of LSD1 and its role in these effects.

6.     In general, it would be nice to see more detail related to the cited clinical trials.  Even if most trials did not result in further clinical advancement of the treatment regimen, they can still identify knowledge gaps and inform future clinical studies.  For example, what are the “notable off-target effects and toxicities” (line 328) of BENSpm, mechanistic basis, strategies to avoid in the future, etc.?

Minor points

Some references are single author et al. and others list many authors

Line 153; typo K-ras

Line 371; analogues should be singular

Lines 374 and 456; check consistency of clinical trial annotations throughout (e.g. Phase I vs. Phase 1)

Line 422 cites ref 129 but Table 4 cites ref 133 for seemingly the same content

Line 544; check grammar

References 11 and 62 are the same

Round 2

Reviewer 1 Report

Comments and Suggestions for Authors

Dear Authors,

We have received the revision of your manuscript entitled "The synergistic benefit of combination strategies targeting tumor cell polyamine homeostasis" (ijms-3069276). After a careful review of the modifications made in response to the reviewers' comments, we are pleased to note that all observations and suggestions have been thoroughly and accurately addressed.

The revision has fully met the required objectives. The updates to the manuscript, including the integration of the most recent information regarding therapeutic strategies based on polyamine metabolism, arginine deprivation, agmatine, nanotechnologies, and specific SMOX inhibitors, significantly enrich the scientific content.

Therefore, we believe that the manuscript is now ready for publication in the special issue of the International Journal of Molecular Sciences on Polyamines in Aging and Disease.

Reviewer 3 Report

Comments and Suggestions for Authors

Dear Authors,

The updates in the manuscript titled "The synergistic benefit of combination strategies targeting tumor cell polyamine homeostasis" (ijms-3069276) significantly enhance the manuscript.

The manuscript is ready for publication.
